



# Non biogenic source is an important but overlooked contributor to aerosol isoprene-derived organosulfates during winter in northern China

Ting Yang[1], Yu Xu[1,2]*, Yu-Chen Wang[3], Yi-Jia Ma[1], Hong-Wei Xiao[1,2], Hao Xiao[1,2], Hua-Yun Xiao[1,2]

[1]School of Agriculture and Biology, Shanghai Jiao Tong University, Shanghai 200240, China

[2]Shanghai Yangtze River Delta Eco-Environmental Change and Management Observation and Research Station, Ministry of Science and Technology, Ministry of Education, Shanghai 200240, China

[3]Division of Environment and Sustainability, Hong Kong University of Science and Technology, Hong Kong SAR 00000, China

*Corresponding authors

Yu Xu

E-mail: xuyu360@sjtu.edu.cn



**Abstract:** Previous measurement-model comparisons of atmospheric isoprene levels
showed a significant unidentified source of isoprene in some northern Chinese cities
during winter. Here, spatial variability in winter aerosol organosulfate (OS) formation
in typical southern (Guangzhou and Kunming) and northern (Xi'an and Taiyuan)
cities, China, was investigated to reveal the influence of potential non biogenic
contributor on aerosol OS pollution levels. Monoterpene-derived OSs were
significantly higher in southern cities than in northern cities, which was attributed to
temperature dependent emission of monoterpenes (i.e., higher temperatures in
southern cities drove more monoterpene emissions). However, isoprene-derived OSs
($OS_i$) showed the opposite trend, with significantly higher levels in northern cities.
Principal component analysis combined with field simulation combustion experiments
suggested that biomass burning rather than gasoline, diesel, and coal combustion
contributed significantly to the abundance of $OS_i$ in northern cities. The comparison
of anthropogenic OS molecular characteristics between particles released from
various combustion sources and ambient aerosol particles suggested that stronger
biomass and fossil fuel combustion activities in northern cities promoted the
formation of more anthropogenic OSs. Overall, this study provides direct molecular
evidence for the first time that non biogenic sources can significantly contribute to the
formation of $OS_i$ in China during winter.

**Keywords:** Aerosol organosulfates, Biogenic precursors, Anthropogenic precursors,
Spatial variation, Influencing factors



## 1. Introduction


Organosulfates (OSs) with a sulfate ester functional group typically contribute 3–
30% of the organic aerosol mass in atmospheric fine particles (PM$_{2.5}$) (Luk´Acs et al.
2009). Moreover, OSs have been estimated to account for up to 12% of the total sulfur
mass in fine particles, playing significant roles in the global biogeochemical cycling
of sulfur (Luk´Acs et al. 2009). In particular, OSs can impact the properties of
aerosols, such as hygroscopicity, acidity, viscosity, and morphology, which are closely
associated with the organic aerosol formation and urban air quality (Riva et al. 2019;
Fleming et al. 2019). Thus, aerosol OSs have attracted significant attention over the
years. However, the mechanisms and key factors impacting the formation and
abundance of aerosol OSs in the real world remain considerable uncertainty, despite
the important insights gained from laboratory simulation experiments (Wang et al.
2021; Yang et al. 2023; Wang et al. 2020).
Previous field studies have indicated that acidity (Duporté et al. 2019), sulfate
(Aoki et al. 2020), aerosol liquid water (Duporté et al. 2016), and oxidants (e.g.,
ozone) (Wang et al. 2021) represent critical factors controlling the formation of OSs
via heterogeneous and liquid phase processes (Brüggemann et al. 2020b). Precursor
emission intensities (e.g., isoprene, monoterpenes, polycyclic aromatic hydrocarbons,
and alkanes) also play an important role in impacting abundance of biogenic and
anthropogenic OSs in ambient aerosols (Wang et al. 2022; Bryant et al. 2021; Yang et
al. 2024). Furthermore, previous studies have identified a large number of CHOS
compounds in smoke particles (e.g., pine branches, corn straw, rice straw, and coal)



(Song et al. 2019; Song et al. 2018; Tang et al. 2020). However, limited studies have
focused on the contribution of different smoke particles to urban aerosol OSs. This
may be an overlooked source of OSs. In general, few field studies have conducted a
comprehensive investigation into the relationship between biogenic and
anthropogenic impacting factors and regional differences in aerosol OS pollution.
This hinders our understanding of the formation and constraints of aerosol OS
pollution in a complex polluted atmospheric environment across diverse cities in
China.
The considerable variations in climatic conditions and air pollution levels in the
northern and southern regions of China during winter (Ding et al. 2014; Ding et al.
2016b) provide a distinctive opportunity to examine the complex influences of
precursors, humidity, acidity, atmospheric oxidants, and anthropogenic pollution on
the formation and abundance of aerosol OSs in the real world (Yang et al. 2024; Yang
et al. 2023; Wang et al. 2021; Hettiyadura et al. 2019). In this study, we conducted the
simultaneous observations of OSs and other chemical components in PM$_{2.5}$ collected
from typical southern (Guangzhou and Kunming) and northern (Xi'an and Taiyuan)
cities in China during winter. Moreover, we also attempted to identify OSs in smoke
particles emitted from combustion of different materials (i.e., rice straw, pine branch,
diesel, gasoline, and coal). The principal aims of this study are 1) to investigate the
spatial differences in aerosol OS pollution in northern and southern China during
winter and 2) to elucidate the key factors that contribute to the spatial variability of
OS pollution, with a focus on the OSs derived from smoke particles.



## 2. Materials and Methods

### 2.1. Site description and sample collection

The research sites are located in four urban areas in China, including Xi'an (XA) Taiyuan (TY), Guangzhou (GZ), and Kunming (KM) (**Figure S1a**). XA and TY are typical northern cities with cold winters (average temperature below 2 ℃ during the study period; **Table S1**). Thus, burning coal and biomass for heating is prevalent in these two cities during winter (Zhou et al. 2017; Ma et al. 2017), which significantly deteriorated the local air quality (**Figure S1b**). GZ and KM represent typical southern cities, with an average air temperature of over 10 ℃ during the winter sampling period (**Table S1**). Clearly, the distinctive climatic conditions in the northern and southern cities during winter may lead to significant spatial differences in the level of air pollution and the emission intensity of biogenic volatile organic compounds (VOCs) (Ding et al. 2014; Xu et al. 2024b).

From 10 December 2017 to 8 January 2018, sampling was performed simultaneously in four cities. Filters contained $PM_{2.5}$ were collected at regular two- to three-day intervals, with the collection duration being 24 hours, using a high-volume air sampler (Series 2031, Laoying, China) at a flow rate of ∼1.05 $m^3$ $min^{-1}$ (Xu et al. 2024a). A blank filter was sampled at each of the study sites. A total of four $PM_{2.5}$ samples were collected and stored at a temperature of −30℃. Meteorological data, including wind speed, relative humidity (RH), and temperature, were obtained from nearby environmental stations. Concurrently, the concentrations of various pollutants, such as $O_3$, $NO_2$, and $SO_2$, were also recorded.



**2.2. Smoke particle collection**


The controlled burning experiments conducted in the field were designed to
simulate the emissions of "real world" burning cases in China (**Figure S2**), with the
methodology being improved according to the previous reports (He et al. 2010; Wang
et al. 2017). Rice straw and pine branch are typical materials for biomass burning in
China (Zhou et al. 2017). In addition, the combustion of coal, gasoline, and diesel was
representative of fossil fuel combustion (Yu et al. 2020). Accordingly, the smoke
particles ($PM_{2.5}$) emitted from rice straw, pine branch, coal combustion, gasoline
vehicle exhausts, and diesel vehicle exhausts were separately collected using self-
made devices.
Briefly, the smoke from the combustion of rice straw, pine branch, and coal was
sampled through a combustion furnace pumped with ambient air (particulate matter is
removed) (**Figure S2a**). It should be noted that introducing ambient air with removed
particulate matter into the combustion furnace is to minimize the pollution of ambient
particulate matter to the smoke particle samples. This is the most distinct difference
from the previous combustion experiment (Zhang et al. 2022; Xu et al. 2023a). Each
combustion experiment for straw, pine branch, and coal lasted for 30−40 min.
Regarding the smoke particles emitted from gasoline vehicle exhausts and diesel
vehicle exhausts, they were collected for 3 hours by directly connecting to the car
exhaust pipe (**Figure S2b**). All smoke particle samples are collected onto prebaked
quartz fiber filters via a high-volume air sampler (Series 2031, Laoying, China). Four
repeated experiments were conducted for each combustion material, one of which was



collected as a blank sample. All smoke particle samples were stored at −30°C.

**2.3. Chemical analysis and predictions of aerosol acidity and water concentration**

The extraction, measurement procedures, and identification of OSs were

described in detail in our recent publications (Yang et al. 2024). Briefly, the filter
sample was extracted using methanol, then filtered through a 0.22 μm PTFE syringe
filter and concentrated by a gentle stream of nitrogen gas. Subsequently, the
concentrated sample with adding ultrapure water (300 μL) was thoroughly mixed
using a mixer. The mixture was centrifuged to obtain the supernatant for analysis of
UPLC-MS/MS system (Waters, USA) (Wang et al. 2021). The reverse-phase liquid
chromatography (RPLC) method was used in this study. Although our method is quite
effective in retaining and separating low molecular weight (MW) OSs, as
demonstrated in our recent publication (Yang et al. 2024), we also acknowledge that
the developed hydrophilic interaction liquid chromatography method may represent a
optimized solution for the measurement of low-MW OSs (Cui et al. 2018; Hettiyadura
et al. 2015).

In addtion, it has been indicated in previous studies (Brüggemann et al. 2020a;

Kristensen et al. 2016) that the levels of OSs can be affected by the sampling
procedure, especially when $SO_2$ removal procedures are not employed. On the
assumption that $SO_2$ reacts with organics on filters to form OSs, similar processes
must also occur on ambient particles prior to sampling. Morover, there is currently no
study evaluating the relative efficiency of OS generation in filters and ambient





particles. Consequently, the possible consequences of sampling without denuding $SO_2$
for the quantification of OSs were not taken into account in our studies (Brüggemann
et al. 2020a; Kristensen et al. 2016). In total, 212 OSs were identified. However, only
111 OS species were quantified using surrogate standards in this study (**Table S2** and
**S3**) (Wang et al. 2021; Hettiyadura et al. 2017). The study divided the several
principal OS groups as follows: monoterpene-derived OSs ($OS_m$), isoprene-derived
OSs ($OS_i$), $C_2-C_3$ OSs (i.e., OSs with two or three carbon atoms), and anthropogenic
OSs (i.e, aliphatic and aromatic OSs) (Yang et al. 2023). The specific classification
and quantification methods were detailed in our recent publications (Yang et al. 2023;
Yang et al. 2024) and **Supporting Information**.

An additional portion of each filter was extracted using ultrapure water for

determining the inorganic ions (Huang et al. 2023). The concentrations of $SO_4^{2-}$, $Ca^{2+}$,
$NO_3^-$, $Na^+$, $K^+$, $Mg^{2+}$, $Cl^-$, and $NH_4^+$ were analyzed using ICS5000+ ion
chromatography (Thermo, USA) (Yang et al. 2024). The mass concentration of
aerosol liquid water (ALW) and pH value were calculated by a thermodynamic model
(ISORROPIA-II) in the forward mode and thermodynamically metastable state, which
was detailed in our previous studies (Liu et al. 2023; Lin et al. 2023; Xu et al. 2022;
Xu et al. 2023b; Xu et al. 2020). The role of OSs in influencing ALW and pH was not
included in this study because their impact on prediction outcomes was deemed to be
insignificant.

**3. Results and Discussion**





**3.1. Spatial variations in concentrations and compositions of different OSs**

**Figure 1a** shows the spatial distributions in mass concentrations and mass fractions of $OS_i$, $OS_m$, aliphatic OSs, aromatic OSs, and $C_2$–$C_3$ OSs in $PM_{2.5}$ collected in southern (KM and GZ) and northern (TY and XA) cities during winter. On average, $OS_i$ was the dominant OS subgroup, which accounted for 37% – 46% and 68% – 69% of the total OS mass in southern and northern cities, respectively. The predominance of $OS_i$ in aerosol OSs was also reported by previous studies in cities in northern (e.g., Beijing and Tianjin) (Wang et al. 2018; Ding et al. 2022) and southern (e.g., Guangzhou and Shanghai) (Wang et al. 2022; Wang et al. 2021) China, as well as in coastal (the Yellow Sea and Bohai Sea) (Wang et al. 2023) and European (Sweden) (Kanellopoulos et al. 2022) and American regions (Chen et al. 2021; Hettiyadura et al. 2017; Hettiyadura et al. 2019) (**Table S4**). Moreover, the concentrations of $OS_i$ were significantly lower in southern cities ($61 \pm 38$ ng m$^{-3}$ – $87 \pm 60$ ng m$^{-3}$) than in northern cities ($171 \pm 69$ ng m$^{-3}$ – $260 \pm 71$ ng m$^{-3}$) (**Table S1**), showing a concentration range overlapped with previous observations (**Table S4**). From southern to northern cities, the mass concentrations and mass fractions of $OS_m$ tended to decrease, which was opposite to the spatial variation pattern of $OS_i$ (**Figure 1a**). Both $OS_i$ and $OS_m$ are generally considered as typical biogenic OSs (Hettiyadura et al. 2019; Wang et al. 2018), the abundances of which were tightly associated with biogenic VOC emissions when acidity, sulfate,  atmospheric oxidation capacity, and ALW are not limiting factors (Bryant et al. 2021; Wang et al. 2022; Yang et al. 2024). Thus, these dissimilarities in the spatial variations of $OS_i$ and $OS_m$ can be attributed to



large differences in the intensity of biogenic VOC emissions (Wang et al. 2022)
and/or the key factors that constrain OS formation between the northern and southern
regions of China (**Table S1**).

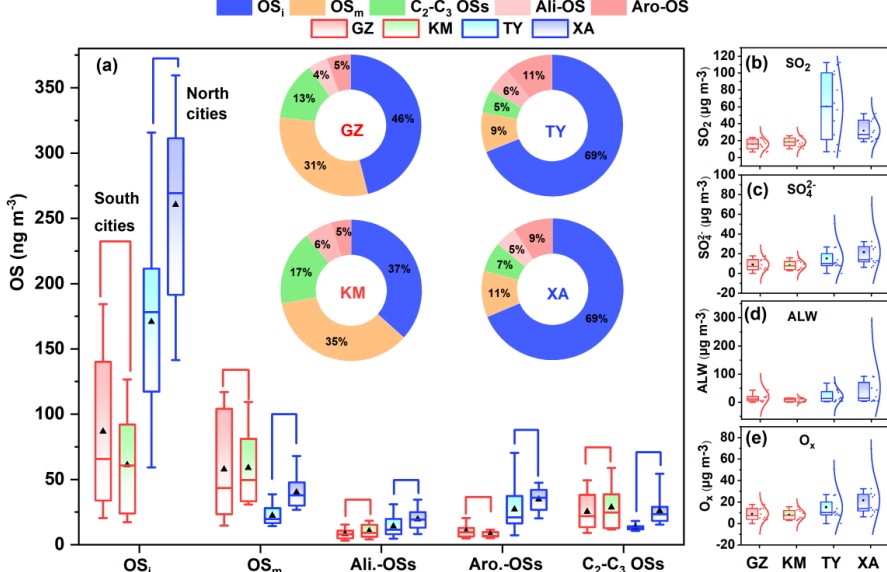

**Figure 1** Box and whisker plots showing the variations in the concentration of
different OS groups in $PM_{2.5}$ collected in southern (GZ and KM) and northern (TY
and XA) cities of China during winter. Each box encompasses the 25th–75th
percentiles. Whiskers are the minimum and maximum values. The triangles and solid
lines inside boxes indicate the mean and median. The spatial variation in average
percentage distributions of various OS groups was shown in panel (a). Spatial
variations in (b) $SO_2$, (c) $SO_4^{2-}$, (d) ALW, and (e) $O_x$ levels.

The abundance of anthropogenic OSs (i.e., $OS_a$, including aliphatic and aromatic
OSs) in southern cities was lower than that of $OS_m$, which was opposite to the case in



the northern cities showing higher anthropogenic OS abundance (**Figure 1a** and
**Table S1**). Moreover, we found that the spatial variation patterns of $OS_i$ and $OS_a$ were
similar to those of $SO_2$, $SO_4^{2-}$, ALW, and $O_x$ (**Figures 1b–e**), as indicated by significant
($P < 0.05$) correlations of $OS_i$ and $OS_a$ with those factors (**Figure S3**). However, $OS_m$
and $C_2$–$C_3$ OSs showed an opposite spatial variation pattern to $SO_2$, $SO_4^{2-}$, ALW, and
$O_x$ (**Figure 1**). If both $OS_i$ and $OS_m$ are assumed to be formed mainly from the
oxidation of biologically emitted VOCs, the higher $SO_2$, $SO_4^{2-}$, ALW, and $O_x$ levels
could theoretically lead to higher $OS_m$ in northern cities, just as these factors leaded to
higher $OS_i$ abundance in northern cities (**Figure 1** and **Figure S3**). Accordingly, the
above differentiated spatial variation patterns among different OS subgroups likely
indicated that other sources of isoprene contributed to the formation of $OS_i$ in
northern cities. Given the significant ($P < 0.05$) correlations between $OS_i$ and $OS_a$
(**Figure S3**), non biogenic isoprene emissions may play an important role in the
formation of aerosol $OS_i$ in northern cities. This will be further demonstrated in the
following discussion.

**3.2. Key factors affecting spatial differences in monoterpene-derived OS**
**abundance**

**Figure 2a** shows the distribution of $OS_m$ concentration as a function of air

temperature. We found that the $OS_m$ concentration tended to increase with the increase
of air temperature. Specifically, the air temperature in the southern cities was mainly
in the range of 7–14°C during the sampling period, corresponding to higher aerosol



$OS_m$ abundance. In contrast, the low temperature (< 7°C) in the northern cities
corresponded to a significant decrease in $OS_m$ abundance. This finding was similar to
the previously observed decrease in aerosol $OS_m$ compounds with decreasing
temperature during winter in Guangzhou (Bryant et al. 2021). Furthermore, the
indicator ($C_L \times C_T$) of biogenic VOC emission rate was also higher in southern cities
than in northern cities (**Figure 2b**), which implied higher monoterpene emissions in
southern cities. It has been suggested that the emission rates of biogenic VOCs (e.g.,
monoterpene and isoprene) can be driven by increased air temperature and lighting
(Ding et al. 2016a; Ding et al. 2016b). A previous study also found that the
concentrations of atmospheric monoterpenes during the winter season were higher in
warmer southern Chinese cities than in colder northern Chinese cities (Ding et al.
2016b; Li et al. 2020). In particular, GZ and KM, which encompass extensive areas of
coniferous and broad-leaved forests, have been identified as hotspots for monoterpene
and isoprene emissions (Li and Xie 2014). Considering the lower levels of key
factors affecting OS formation observed in southern cities (**Figures 1b–e** and **Table**
**S1**), it can be inferred that the significant spatial differences in $OS_m$ abundances were
largely attributed to temperature dependent emission of monoterpenes.



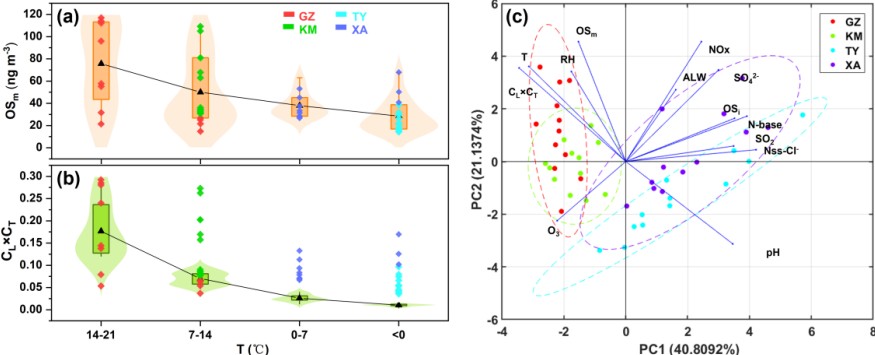

**Figure 2** Distribution of (a) $OS_m$ and (b) $C_L \times C_T$ data in different temperature ranges during winter. The triangles inside boxes indicate the mean. Principal component analysis result (c) deciphering the relationship among $OS_i$, $OS_m$, and key factors influencing OS formation.

To further determine the key factors affecting the spatial differences of $OS_m$, principal component analysis was conducted (**Figure 2c**). It can be easily determined that the abundance of aerosol $OS_m$ was closely related to changes in air temperature and $C_L \times C_T$ value. This precisely explained the changes in $OS_m$ data in the southern cities. In contrast, the abundance of aerosol $OS_i$ in the northern cities was more influenced by anthropogenic factors, as indicated by combustion source tracers such as nitrogen-containing bases (N-bases) and non-sea-salt $Cl^-$ (nss-$Cl^-$) (Wang et al. 2017; Jiang et al. 2023) (**Figure 2c**). Thus, principal component analysis can perfectly distinguish the main factors causing changes in $OS_m$ and $OS_i$ abundances between the northern and southern cities. In general, the above results confirm that the spatial variation of $OS_m$ was predominantly controlled by temperature-related monoterpene



emissions. However, this cannot fully account for the observed spatial variation of
$OS_i$. Interestingly, the spatial distribution patterns of $OS_m$ and $OS_i$ in northern and
southern China exhibited consistency during summer, closely resembling the spatial
distribution of biogenic VOC emission intensities (Wang et al. 2022). Thus, this case
together with our observations during winter further imply that non biogenic sources
of isoprene were important contributors to the formation of $OS_i$ in northern China
during winter.

**3.3. Significant contribution of biomass burning to isoprene-derived OSs in**
**Northern China**

The previous principal component analysis has suggested that the abundance of

$OS_i$ in northern cities was closely related to the levels of combustion source tracers
(e.g., N-base compounds and nss-Cl$^-$). To further confirm the potential contribution of
combustion release to aerosol $OS_i$, OSs in smoke particles ($PM_{2.5}$) emitted from rice
straw, pine branch, and coal combustion, as well as from gasoline vehicle exhausts,
and diesel vehicle exhausts, were investigated. A total of 8 distinct $OS_i$ were identified
in both the smoke particles emitted from biomass burning (rice straw and pine branch)
and the ambient aerosol particles, including $C_4H_7O_6S^-$, $C_5H_9O_6S^-$, $C_5H_{11}O_6S^-$,
$C_5H_7O_7S^-$, $C_4H_7O_5S^-$, $C_5H_{11}O_7S^-$, $C_5H_9O_7S^-$, and $C_5H_9O_8S^-$. Moreover, the peak
intensities of these 8 $OS_i$ in smoke particles emitted from fossil fuel combustion
(gasoline and diesel vehicle exhausts and coal) were close to those in the blank
sample. A previous investigation into CHOS compounds in smoke particles emitted



from residential coal combustion and biomass burning also failed to identify $OS_i$
species (Song et al. 2019; Song et al. 2018), which further supported the reliability of
the combustion experiment conducted in this study. $C_5H_9O_6S^-$ was dominant $OS_i$
species in pine-derived smoke particles (**Figure 3a,c**). We found that the average
concentration of $C_5H_9O_6S^-$ in ambient aerosol samples was much higher in northern
cities than in southern cities (**Figure 3b**). A reasonable explanation for this is that pine
branches are commonly used as solid fuel for heating and cooking in northern suburbs
and rural areas (Zhou et al. 2017). $C_5H_7O_7S^-$ and $C_4H_7O_5S^-$ dominated $OS_i$ species in
straw-derived smoke particles (**Figure 3a,c**). However, these two types of $OS_i$ have
relatively low abundance in ambient aerosol samples in both northern and southern
cities. This may be attributed to the fact that straw burning was mainly concentrated
in autumn rather than winter in China (Zhou et al. 2017; Yang et al. 2015). On
average, the biomass burning-related $OS_i$ accounted for 58% − 64% and 86% − 87%
of the total $OS_i$ concentration in southern and northern cities, respectively (**Figure 3c**).
Although these biomass burning-related $OS_i$ can also be formed through atmospheric
transformation of biogenic isoprene, the higher proportion of these $OS_i$ in northern
cities together with previous principal component analysis results still support our
previous consideration that non biogenic $OS_i$ may be an important contributor to
aerosol $OS_i$ in northern cities.





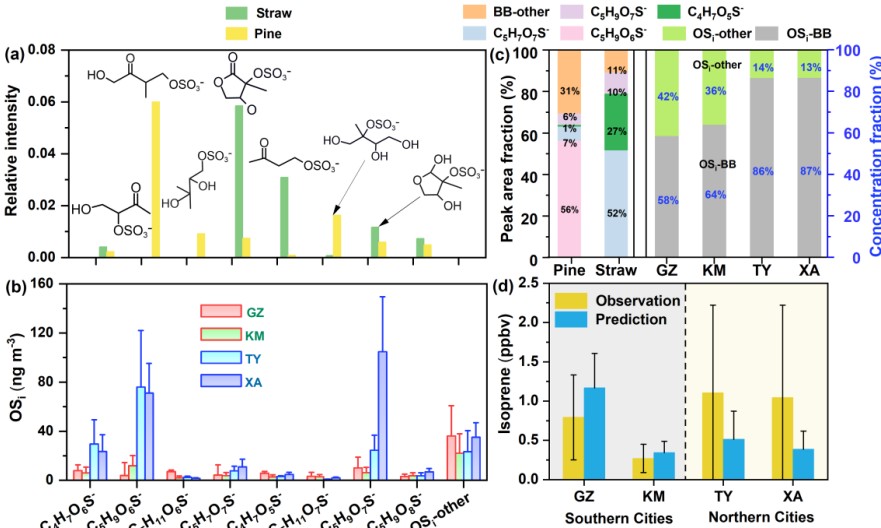

**Figure 3** Relative signal intensity of (a) identified major $OS_i$ species in different types

of smoke particle samples. Spatial variation in the concentration of $OS_i$ identified in

smoke particles in (b) ambient $PM_{2.5}$ samples. Peak area and concentration fraction of

(c) $OS_i$ species identified in both ambient $PM_{2.5}$ samples collected in different cities

and smoke particles. Comparison of (d) isoprene mixing ratios obtained from

observation and modeling in different cities (Zhang et al. 2020).

Previous laboratory studies have suggested that these identified $OS_i$ species in

biomass burning-derived smoke particles are typically formed through heterogeneous

and multiphase reactions associated with isoprene, its oxidation intermediates, and

sulfate or sulfur dioxide (Surratt et al. 2008; Surratt et al. 2007; Darer et al. 2011).

Specifically, $C_5H_9O_6S^-$, as a sulfate ester of $C_5$-alkene triols, was formed mainly

through the uptake of gas-phase isoprene oxidation products onto acidified sulfate

aerosol (Surratt et al. 2007). The formation of $C_5H_7O_7S^-$ and $C_5H_9O_7S^-$ begins with



the gas-phase oxidation of isoprene (Surratt et al. 2008). $C_4H_7O_6S^-$ can be generated
both from isoprene photooxidation and sulfate radical reaction with methacrolein
(MACR) or methyl vinyl ketone (MVK) (Schindelka et al. 2013; Wach et al. 2019;
Nozière et al. 2010). $C_5H_{11}O_7S^-$ was produced by reactive uptake of isoprene-derived
epoxide (IEPOX) on sulfate under low-NOx conditions. Since our combustion
experiments have excluded the direct contribution of ambient aerosol particles to $OS_i$
in smoke particles, it can be expected that these detected $OS_i$ compounds were mainly
generated within smoke plumes through the isoprene oxidation pathway mentioned
above. It has been demonstrated that directly emitted organic aerosols or VOCs can
undergo a chemical reaction within smoke plumes, forming secondary organic
compounds within a matter of hours (Wang et al. 2017; Song et al. 2018; Mason et al.
2001). A field study conducted by Zhu et al. (2016) at a rural site (Yucheng) in the
North China Plain (NCP) region has observed that the concentration of ambient
isoprene during the period of straw combustion was approximately twice as high as
that observed during periods of non combustion. In addition, Li et al. (2018) found
that isoprene-derived epoxides increased significantly during field open burning of
straw. Generally, despite the fact that a few of the mechanisms by which OSs are
formed have been verified through field studies, the formation of CHOS and CHONS
compounds has been observed to occur in the biomass burning plume (Zhang et al.
2024; Song et al. 2018; Tang et al. 2020). Thus, these previous case studies further
support our consideration that $OS_i$ compounds formed in biomass burning-derived
smoke particles in this study can be attributed to increasing isoprene emission caused



by field biomass burning (Zhu et al. 2016) and favorable aqueous secondary organic
aerosols (SOA) formation during the aging process of the biomass burning plume
(Gilardoni et al. 2016).
**Figure 3d** presents a comparison between the isoprene mixing ratios derived
from model simulations (plant functional type related model) and those observed in
the field in different Chinese cities during winter (December and January) (Zhang et
al. 2020). Overall, the levels of isoprene observed in northern cities during winter
were higher than those in southern cities. In addition, the predicted values in southern
cities were slightly higher than the observed values, which may be attributed to the
lag in model prediction results caused by the rapid urbanization rates in these southern
cities (Zhang et al. 2020). However, the observed values in these two northern cities
were 53% to 63% higher than the predicted values, on average. Clearly, this plant
functional type related isoprene prediction model cannot explain the large amount of
"missing" isoprene sources in northern cities. Thus, the observed spatial differences in
$OS_i$ (**Figure 1**) and field combustion experiments (**Figure 3**) can suggest that these
"missing" isoprene sources were mainly derived from biomass burning, significantly
contributing to the production of aerosol $OS_i$ in northern cities. This can be also
supported by previous principal component analysis and correlation analysis among
combustion source tracers and $OS_i$ species (**Figure 2** and **Figure S4**).

**3.4. Formation of anthropogenic OSs mainly driven by fossil fuel and biomass**
**combustion**



**Figures 4a,b** show the average concentration distribution of anthropogenic OSs
classified based on the number of O atoms in their molecules in southern (GZ and
KM) and northern (TY and XA) cities. The $O_4S_1$ subgroup was the most abundant
aromatic OSs in both southern and northern cities, among which $C_9H_9O_4S^-$, phenyl
sulfate ($C_6H_5O_4S^-$), and benzyl sulfate ($C_7H_7O_4S^-$) were dominant species (**Table S3**).
$C_7H_7O_4S^-$ and $C_6H_5O_4S^-$ have been suggested to be formed mainly through the
photooxidation of 2-methylnaphthalene and naphthalene (Riva et al. 2015), or
alternatively, by the sulfate radical reaction with aromatic compounds, including
toluene and benzoic acid, in an aqueous phase environment (Riva et al. 2015). The
formation mechanism of $C_9H_9O_4S^-$ is rarely reported. However, $C_9H_9O_4S^-$, $C_6H_5O_4S^-$,
and $C_7H_7O_4S^-$ were also detected in both fossil fuel combustion-derived smoke
particles and biomass burning-derived smoke particles (**Figure S5** and **Table S5**),
indicating that the aromatic VOCs produced by fuel combustion are closely related to
the formation of these aromatic OSs. Overall, aerosol aromatic OS compounds in both
southern and northern cities were mainly distributed between four and six O atoms
(**Figure 4c**), which was similar to the distribution of aromatic OSs identified in
various smoke particles emitted from different combustion sources (**Figure 4d**).
However, the average abundances of aromatic $O_{4-6}S_1$ compounds in northern cities
were 3–6 times higher than those in southern cities. The above results suggest that
aromatic OSs originated from fossil fuel and biomass combustion activities are
important contributors to urban aerosol anthropogenic OSs in winter in China,
especially in northern cities. We found that the correlations between aromatic OSs and





anthropogenic indicators ($SO_2$, $SO_4^{2-}$, N-base, and nss-$Cl^-$) were stronger in northern
cities than in southern cities (**Figure S6**), and that the release of polycyclic aromatic
hydrocarbons from fossil fuel combustion was also higher in northern cities (**Figure**
**S7**). This further indicates that higher aerosol aromatic OSs in northern cities was
mainly attributed to stronger combustion activities in those cities.

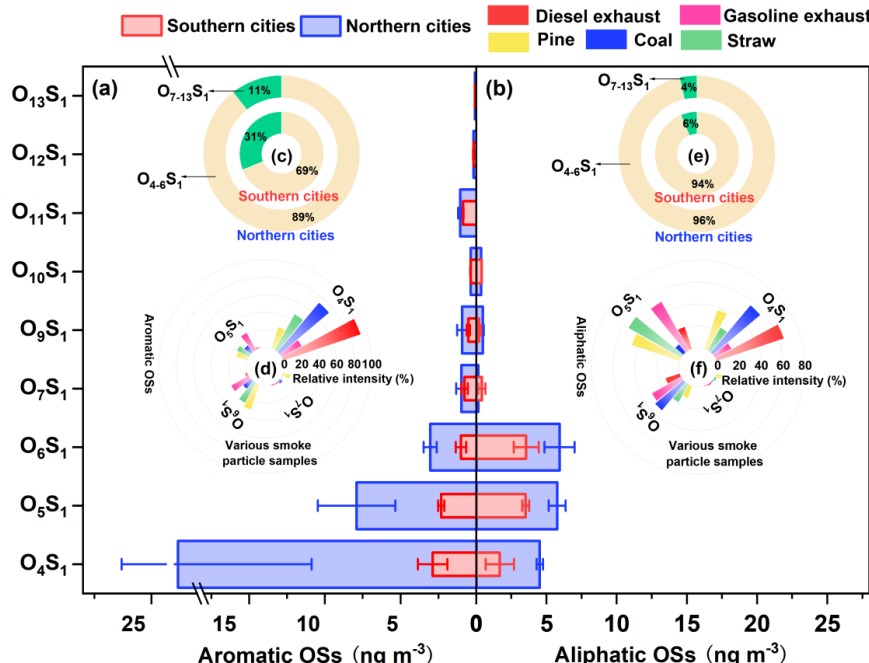

**Figure 4** Concentration distribution of different (a) aromatic and (b) aliphatic OS
subgroups (classification based on oxygen atoms) in southern and northern cities.
Ring charts (c,e) show the percentage contributions of $O_{4-6}S_1$ and $O_{7-13}S_1$ subgroups.
Radial bar charts (d,f) illustrate the relative signal intensity of different OS subgroups
in different smoke particle samples.

Aliphatic OSs were also predominantly distributed between $O_4S_1$ and $O_6S_1$





subgroups in both southern and northern cities (**Figures 4b,e**), which was similar to
the case found in both fossil fuel combustion-derived smoke particles and biomass
burning-derived smoke particles (**Figure 4f**). It has been suggested that the long-chain
alkanes derived from traffic emissions can largely contribute to the formation of
CHOS compounds with aliphatic carbon chains (Tao et al. 2014). In addition, Tang et
al. (2020) analyzed the molecular compositions of smoke particles from open biomass
burning, household coal combustion and vehicle emissions and suggested that the
aliphatic CHOS compounds can be derived from both vehicle emissions and coal and
biomass combustion. In this study, aliphatic OSs showed a significant ($P < 0.05$)
positive correlation with nss-Cl$^-$, SO$_2$, NO$_x$, and N-base compounds in both southern
and northern cities (**Figure S8**), indicating aerosol aliphatic OSs were affected by a
combination of biomass burning and vehicle emissions in those cities during winter.
Thus, the significantly higher level of aliphatic O$_{4-6}$S$_1$ species in northern cities
indicated that the formation of aliphatic OSs in northern cities was more driven by
pollutants released from the combustion of fossil fuels and biomass compared to
southern cities. This consideration is highly consistent with the fact that the
concentrations of air pollutants (e.g., SO$_2$ and NO$_2$) in northern cities with a large
demand for heating during winter are usually higher than those in warmer southern
cities (**Figure S1b**) (Yu et al. 2020; Ding et al. 2017; Ma et al. 2017; Zhou et al.

2017).


**4. Conclusion and atmospheric implications**



It has been previously suggested that isoprene can also be released into the
atmosphere as a result of open burning of agricultural residues and forest fires
(Andreae 2019; Simpson et al. 2011). A field study conducted by Wang et al. (2019)
in Beijing during winter inferred that the prevalence of $OS_i$ compounds in total
aerosol OSs may be partially attributable to biomass burning emissions, although
there was a paucity of compelling evidence to support this hypothesis. This work
combines strongly contrasting observational studies (northern Chinese Cities vs
southern Chinese Cities) with in situ combustion modelling experiments to provide
the first direct evidence that biomass burning emission, rather than fossil fuel
combustion emission, is a significant contributor to aerosol $OS_i$ in northern cities
(**Figure 5**). In Chinese cities, particularly those in the northern region, biomass
materials are extensively utilized for domestic heating and cooking purposes during
the winter season (Zhou et al. 2017). Clearly, the isoprene emissions from biomass
combustion sources would result in higher isoprene mixing ratios than those
simulated by the model (Zhang et al. 2020) that only considers natural isoprene
emissions. Thus, isoprene prediction models applied to Chinese winters in the future
should also take into account the various biomass combustion source releases.
Furthermore, biogenic OSs are important SOA constituents and have been frequently
serve as important tracers for biogenic SOA (Ding et al. 2014; Ding et al. 2016a). The
overall results suggest that some $OS_i$ species may not be suitable as biogenic SOA
markers, especially in areas with intensive biomass burning activities, such as
northern Chinese cities during winter.



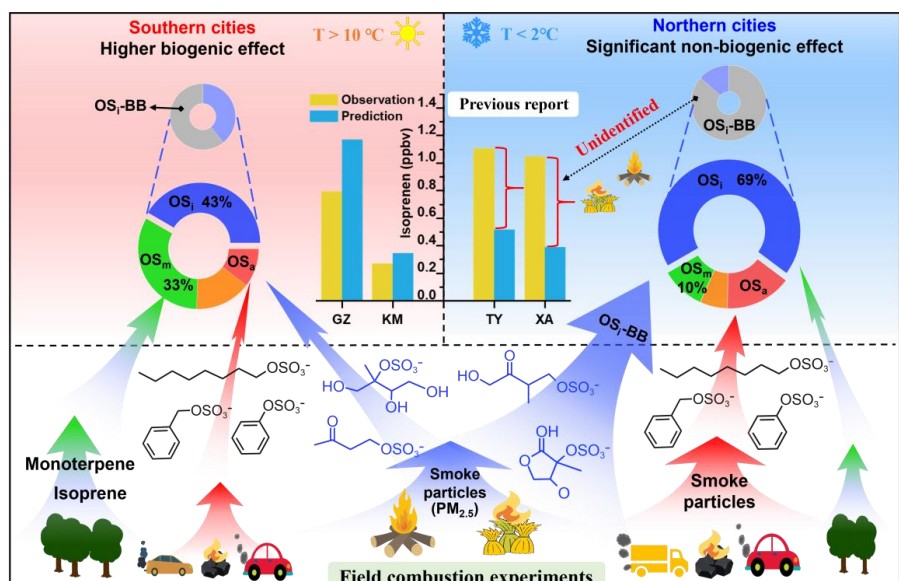


**Figure 5** Conceptual picture showing the characteristics and main contributors of OSs
in northern and southern China during winter.

We found that different fossil fuel combustion emissions (e.g., vehicle emissions
and coal combustion emissions) and biomass burning emissions can contribute to
aerosol anthropogenic OSs. However, current studies have not been able to accurately
distinguish between the contributions of various material combustion to different
types of anthropogenic OSs. Future research is necessary to develop more
comprehensive models to further explore the effects of various combustion sources on
the generation and reduction of urban aerosol OS pollution. Of particular importance
is that although the production of various OSs was directly observed through our
simulated combustion experiments, it is not clear whether the chemical mechanisms
involved are similar to those derived from the laboratory simulations. This is because



462 the combustion process is accompanied by the effects of high temperatures. In

463 general, although our results provide direct evidence for the release of OSs from

464 combustion of various combustion sources, further mechanistic studies and

465 environmental impact assessment are still urgently needed. This may be important for

466 effective control of urban wintertime organic aerosol pollution in China.

468 **Data availability**

469 The data presented in this work are available upon request from the corresponding

470 authors.

472 **Competing interests**

473 The authors declare no conflicts of interest relevant to this study.

475 **Supplement**

476 Additional information regarding descriptions of classification of OSs (Text S1 and

477 Table S2–S3), quantification of OSs (Text S2), estimating of isoprene emission rate

478 (Text S3), the OS concentrations showed in previous studies (Table S4), the location

479 of the sampling site (Figure S1), smoke particle collection (Figure S2), and more data

480 (Table S1, Table S5, and Figures S3−S8).


482 **Author contributions**

483 YX designed the study. TY, YJM, HWX, and HX performed field measurements and



sample collection; TY and YJM performed chemical analysis; YX and TY performed
data analysis; YX and TY wrote the original manuscript; and YX, TY, YCW, and
HYX reviewed and edited the manuscript.

**Acknowledgements**
The authors are very grateful to Jian-Zhen Yu at the Hong Kong University of Science
and Technology for her kind and valuable comments to improve the paper.

**Financial support**
This study was kindly supported by the National Natural Science Foundation of China
(grant number 42303081 and 22306059), the Shanghai "Science and Technology
Innovation Action Plan" Shanghai Sailing Program (grant number 22YF1418700),
and the National Key Research and Development Program of China (grant number
2023YFF0806001).





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
