# Peer review of "Non biogenic source is an important but"

_EGUsphere, 2024_

## Author Comment (AC1)

**General.**

We would like to express our sincere appreciation to the editor and reviewers for their valuable feedback and constructive suggestions, which significantly improved the manuscript. We have carefully addressed all the reviewers' concerns and made the revisions. Responses to specific comments raised by the reviewers are described below. All changes made in the revised manuscript are highlighted in red, and our detailed responses to the specific comments are presented below in blue.

**Comments of Referee #1 and our responses to them**

Comments:

*This study investigated the spatial variations of the concentration of different types of organosulfates in cities in southern and northern China. The manuscript further analyzed the factors leading to significant regional differences in organosulfate concentration and composition between the southern and northern cities. Based on principal component analysis, correlation analysis, organosulfate formation mechanism analysis, and field simulation combustion experiments, the authors concluded that the emissions from biomass burning, rather than those from gasoline, diesel, and coal combustion, can play a significant role in the formation of isoprene-derived organosulfates. This is a significant and valuable finding. It thus follows that when attempting to predict winter isoprene emissions in China, it is essential that the influence of biomass combustion emissions is duly taken into account, with due consideration given to the regional differences in question. Overall, the manuscript presents high quality research data based on field observations and simulation experiments. The content presented here will be of interest to the readership of Atmospheric Chemistry and Physics. After the authors address the following minor comments, this manuscript would be suitable for publication.*

Response: We are very grateful for your professional and thoughtful review of our manuscript. We have revised the manuscript to address the comments. Our responses to the specific comments and changes made in the manuscript are given below.

Specific comments:

1) *Lines 142-143: I suggest that the authors add specific information about the columns used here or in the supplemental file. It is important to consider this when comparing the analysis method from this study with those from previous or future research.*

Response: Thank you for your suggestion. The T3 column was used in this study, and this information has been incorporated into the manuscript.

Lines 138-140: …The reverse-phase liquid chromatography (RPLC) method was performed on an Acquity UPLC HSS T3 column (2.1mm × 100 mm, 1.8 µm particle size; Waters, USA) in this study.…

2) *Lines 157-158: Why did the authors choose 111 of the OSs for further quantification? I have checked your previously published paper showing quantification of 106 OS species. Is there any difference here? Please clarify.*

Response: Thank you for your question. In our previous study, we quantified 106 species OSs species, which represented the species detected in the $PM_{2.5}$ samples at Urumqi. However, in this study, we analyzed samples from four different cities, thereby facilitating the capture of a more extensive array of OSs. Consequently, we were able to detect additional OS species, resulting in a greater number of species being quantified. This explains why we selected 111 OSs for further quantification in this study.

3) *Lines 172-174: How is this insignificance determined or evaluated?*

Response: Thank you for your comment. In our previous study, the different outputs of the pH values between the cases without and with including OSs as additional sulfates were compared to examine the potential influence of OSs on pH calculation (Yang et al. 2024). An insignificant difference was found between these two predictions, suggesting a negligible contribution of OSs to pH value. In addition, the aerosol pH may be overestimated by 0, 3.5, and 9.5 units when the ratios of total organic sulfur to total inorganic sulfur were 0, 0.73, and 4.66, respectively (Riva et al. 2019). The ratio of the total OSs to total sulfates in Guangzhou, Kunming, Taiyuan, and Xi'an was 0.019, 0.022, 0.017 and 0.021, respectively, further indicating that the effect of OSs on acidity prediction was neglected in this study.

Lines 171-174: The influence of OSs on ALW and pH was not taken into account in the present study due to their negligible contribution to the prediction outcomes, as indicated by Riva et al. (2019) and Yang et al. (2024).

4) *Line 196: Please remove any possible extra spaces between 'sulfate' and 'atmospheric oxidation capacity'.*

Response: Thank you for your comment. We have deleted extra spaces between 'sulfate' and 'atmospheric oxidation capacity'.

Line 195: …"acidity, sulfate, atmospheric oxidation capacity"…

5) *Line 217: Please change 'an' to 'the'.*

Response: The revision has been made (Line 216).

Line 216: …"showed the opposite spatial variation pattern"…

6) *Lines 224-225: Was the correlation analysis presented in Figure S3 using all the data from this study? In addition, I suggest adding 'further' before 'given'.*

Response: Thank you for your insightful comment. The correlation analysis presented in **Figure S3** used all the data in this study. In addition, we have added 'further' before 'given' in Line 230.

Line 223: …"Further given the significant"…
**SI Figure S3**: …(using data from four cities)…

7) *Line 248: Please remove any possible extra spaces before 'considering'.*

Response: Thank you for your comment. The revision has been made.

Line 248: …"Considering the lower levels"…

8) *Line 261: Please change ' precisely' to 'further'.*

Response: The revision has been made.

Line 261: …"This further explained the changes"...

9) *Line 269: I suggest the author delete 'fully' or replace it with a more appropriate word.*

Response: The revision has been made.

Line 269: "However, this cannot account for the observed spatial variation of $OS_i$"

10) *I know that those isoprene-derived organosulfates were detected in the samples*

*from the simulated combustion experiments, however, I would like to know how you think about their formation. This could be a critical question for future research in this area, although it does not affect the results of this study.*

Response: Thank you for your valuable comment. In this study, the formation of $OS_i$ can be attributed to the oxidation of isoprene in smoke plumes. Previous experimental studies have shown that these $OS_i$ compounds typically form through the heterogeneous and multiphase reactions of isoprene and its oxidation intermediates with sulfate or sulfur dioxide (Surratt et al. 2008; Surratt et al. 2007; Darer et al. 2011). Since our combustion experiments have excluded the direct contribution of ambient aerosol particles to $OS_i$ in smoke particles, it can be expected that these detected $OS_i$ compounds were mainly generated within smoke plumes through the isoprene oxidation pathway mentioned above. It has been demonstrated that directly emitted organic aerosols or VOCs can undergo a chemical reaction within smoke plumes, forming secondary organic compounds within a matter of hours (Wang et al. 2017; Song et al. 2018; Mason et al. 2001). Despite the fact that a few of the mechanisms by which OSs are formed have been verified through field studies, the formation of CHOS and CHONS compounds has been observed to occur in the biomass burning plume (Zhang et al. 2024; Song et al. 2018; Tang et al. 2020). Thus, these previous case studies further support our consideration that $OS_i$ compounds formed in biomass burning-derived smoke particles in this study can be attributed to increasing isoprene emission caused by field biomass burning (Zhu et al. 2016) and favorable aqueous secondary organic aerosols (SOA) formation during the aging process of the biomass burning plume (Gilardoni et al. 2016).

More discussions are presented in Lines 321–352 in the revised manuscript.

11) *I suggest the authors add a space after Ali. in Figure S3. Furthermore, how was the relative intensity calculated in Figure S5? Please clarify it.*

Response: Thank you for your comment. **Figure S3** has been updated. The relative signal intensity refers to the percentage of the target OS signal intensity in the total signal intensity of the OS group to which the target OS belongs (**Figure S6**).

[Figure]

**Figure S3**. Diagrams presenting Pearson correlations among the concentrations of $O_x$, $SO_2$, $SO_4^{2-}$, and the different OSs (using data from four cities). The numbers in the matrix refer to the correlation coefficients ($r$). Symbols * and ** indicate $P < 0.05$ and $P < 0.01$, respectively.

**Figure S6**. Mean relative signal intensities of typical aromatic OSs (i.e., $C_6H_5O_4S^-$, $C_7H_7O_4S^-$, $C_8H_7O_4S^-$, and $C_9H_9O_4S^-$) in different smoke particle samples. The relative signal intensity refers to the percentage of the target OS signal intensity in the total signal intensity of the OS group to which the target OS belongs.

**At last, we deeply appreciate the time and effort you've spent in reviewing our manuscript.**

**Reference:**

[revised manuscript text omitted]

---

## Author Comment (AC2)

**General.**

We would like to appreciate the editor and reviewers for providing the valuable comments and a better perspective on our work to improve the manuscript. We have revised our manuscript by fully taking the reviewers' comments into account. Responses to specific comments raised by the reviewers are described below. All the changes made and appeared in the revised text are shown in red. All detailed answers to comments are displayed in blue.

**Comments of Referee #2 and our responses to them**

Comments:

*The present study employed a systematic approach to characterize organosulfates in $PM_{2.5}$ samples from both southern and northern Chinese cities during the winter months, complemented by the analysis of smoke particle samples obtained from simulated combustion experiments. The analysis indicated that biomass burning, in contradistinction to the combustion of gasoline, diesel, and coal, exerted a significant influence on the increased levels of particulate isoprene-derived organosulfates observed in northern urban areas. The authors pointed out that stronger biomass and fossil fuel combustion activities in the northern cities resulted in the formation of a greater number of anthropogenic organosulfates. In general, this work offers a compelling example and novel insights into understanding organosulfate pollution in Chinese cities. The manuscript is well organized and its topic is very interesting. Thus, I think it can be accepted after a minor revision.*

Response: We sincerely appreciate your professional and constructive review of our manuscript. Your valuable feedback has greatly improved the clarity and quality of our work.

Detailed comments:

1) *Lines 72-74: It is recommended that the author rephrase the sentence to enhance clarity. For example: This complicates our understanding of how aerosol OS pollution is formed and what limits it in a complex polluted atmosphere across different cities in China.*

Response: Thank you for your suggestion. We have rephrased the sentence.

Lines 70-71: …This complicates our understanding of how aerosol OS pollution is formed and what limits it in a complex polluted atmosphere across different cities in China…

2) *Line 102-107: it seems to me that about ten samples were collected in each city, so at least 40 samples were studied by this study, why here stated " a total of four $PM_{2.5}$ samples were collected and stored at a.....? please clarify.*

Response: Thank you very much for your careful review. We apologize for the confusion caused by this. A total of 12 samples were collected from each city. Thus, we collected a total of 48 samples in four cities. The revision has been made in the revised manuscript.

Line 104: …A total of 48 ambient samples…

3) *Line 146: Please change 'a optimized solution' to 'an optimized solution'.*

Response: Thank you for your comment. The sentence has been rephrased in the revised manuscript.

Line 142-144: …we also acknowledge that the developed hydrophilic interaction liquid chromatography method may provide another solution for the measurement of low-MW OSs…

4) *Line149-150: the authors mentioned that the two references (Brüggemann et al. 2020a; Kristensen et al. 2016) emphasize the impact of the sampling process on the quantitative results of OSs. However, Lines 156-157: the authors also mentioned that the possible consequences of sampling without denuding $SO_2$ for the quantification of OSs were not taken into account in our studies (Brüggemann et al. 2020a; Kristensen et al. 2016). It is strange to quote the same reference in sentences with different meanings. Please clarify.*

Response: We apologize for the incorrect citation. We have clarified the issue and corrected the references accordingly.

Lines 152-154: …Consequently, the possible consequences of sampling without denuding $SO_2$ for the quantification of OSs were not taken into account in our studies (Yang et al. 2023; Yang et al. 2024)…

5) *Line 158: What are the surrogate standards? How were they obtained? What are the recoveries of the surrogated standards?*

Response: We appreciate your valuable comments on our work. To address the comments, the Supporting Information has been updated with additional content.

**Supporting Information**

**S2. Quantification of OSs**

It is evident that OSs with similar carbon backbone structures typically exhibit analogous MS responses (Wang et al. 2021). Consequently, the selection of a surrogate standard for a specific OS was predominantly contingent on the similarity between the carbon chain structures of the targeted OS species and the OS standard (Hettiyadura et al. 2017). Furthermore, the sulfur-containing fragment ions observed in the MS/MS spectra of the standard and targeted OS species have been taken into

consideration (Hettiyadura et al. 2019; Bryant et al. 2021). The recoveries of the aforementioned surrogate standards were, in order, 88%, 84%, 94%, 89%, 88%, 87%, and 84%. Additional details on the identification of OS compounds, their classification and quantifacation, and data quality control are available in our recent publications (Yang et al. 2023; Yang et al. 2024).

6) *Line 159-164: here I suggest to give a brief description on how $OS_m$ and $OS_i$ were defined, which is helpful for readers to quickly understand what the author have done in this study, although more details can be found in **SI**.*

Response: Thank you for the suggestion. The terms "$OS_m$" and "$OS_i$" refer to organosulfates generated from monoterpenes and isoprene, respectively. These compounds were generally classified as biogenic OSs due to their natural origin (Wang et al. 2021; Wang et al. 2018).

Lines 159-161: The terms "$OS_m$" and "$OS_i$" refer to organosulfates generated from monoterpenes and isoprene, respectively. These compounds were generally classified as biogenic OSs due to their natural origin (Wang et al. 2021; Wang et al. 2018).

7) *Line 159-164, Line 167: Please remove any extra spaces between $K^+$ and $Mg^{2+}$.*

Response: We appreciate your attention to detail. The revision has been made.

Lines 165-166: …The concentrations of $SO_4^{2-}$, $Ca^{2+}$, $NO_3^-$, $Na^+$, $K^+$, $Mg^{2+}$, $Cl^-$, and $NH_4^+$…

8) *Line 172-174: here should give a reference to support this conclusion.*

Response: Thank you for your comment. We have added some references to support this.

Lines 171-174: The influence of OSs on ALW and pH was not taken into account in the present study due to their negligible contribution to the prediction outcomes, as indicated by Riva et al. (2019) and Yang et al. (2024).

9) *Line 233: Does the spatial variation of OS$_i$ concentration have temperature dependence?*

Response: Thank you for your comment. A new Figure (as **Figure S4**) has been added to demonstrate that the spatial variation of OS$_i$ was not temperature-dependent.

[Figure]

**Figure S4** Spatial variation of OS$_i$ concentration and temperature (T).

Lines 269-270: However, this cannot account for the observed spatial variation of OS$_i$ (**Figure 2c** and **Figure S4**).

10) *Line 239: For the statement 'indicator ($C_L \times C_T$) of biogenic VOC emission rate',*

*please add some references to support it.*

Response: The references have been added in the revised manuscript.

Lines 237-239: …Furthermore, the indicator ($C_L \times C_T$) of biogenic VOC emission rate (Ding et al. 2016; Guenther et al. 1993) was also higher in southern cities than in northern cities (**Figure 2b**)…

11) *Line 281: what are the "N-base compounds" ? please give more explanation.*

Response: Thank you for your valuable comment. N-base compounds are CHN species that contain exclusively C, H, and N atoms, and have been demonstrated to exhibit high sensitivity as molecular indicators in identifying biomass burning (Wang et al. 2017).

Lines 281-283: …N-base compounds are CHN species that contain exclusively C, H, and N atoms, and have been demonstrated to exhibit high sensitivity as molecular indicators in identifying biomass burning (Wang et al. 2017).…

12) *Section 3.3: Are the $OS_i$ species detected in smoke particles directly emitted or are they produced secondarily?*

Response: Thank you for your valuable comment. The $OS_i$ species detected in smoke particles may not be directly emitted but are produced secondarily. The formation of these compounds is predominantly initiated by the oxidation of isoprene, followed by complex reactions within the smoke plume (Wang et al. 2017; Song et al. 2018; Mason et al. 2001).

More discussions are presented in Lines 321–352 in the revised manuscript.

13) *Figure 5: Based on my understanding, OS$_i$-BB can not only come from biomass combustion, but also from atmospheric transformation of isoprene derived from biological sources. Therefore, although these OS species were indeed detected in the particulate matter released from biomass burning, in order to avoid misleading readers into thinking that these OS species were all from biomass burning release, I recommend the author to add relevant explanations in the caption.*

Response: Thank you for your valuable feedback. To ensure that readers are not misled into thinking that these OS species are solely from biomass burning emissions, the relevant explanation has been added in the figure caption.

Lines 455-457: It is noteworthy that OS$_i$-BB can originate not only from biomass combustion, but also from the secondary formation of isoprene emitted from biogenic sources.

**At last, we deeply appreciate the time and effort you've spent in reviewing our manuscript.**

**Reference**

[revised manuscript text omitted]

---

## Author Comment (AC3)

**General.**

We would like to appreciate the editor and reviewers for providing the valuable comments and a better perspective on our work to improve the manuscript. We have revised our manuscript by fully taking the reviewers' comments into account. Responses to specific comments raised by the reviewers are described below. All the changes made and appeared in the revised text are shown in red. All detailed answers to comments are displayed in blue.

**Comments of Referee #3 and our responses to them**

Comments:

*The primary objective of this study is to elucidate the role of non-biogenic emissions (e.g., rice straw, pine branch, gasoline, diesel, and coal combustion) in the formation of isoprene-derived organosulfates in aerosols in China during winter. The authors synthesized data from large-scale observational studies (comparing northern and southern Chinese cities) with data from simulated combustion experiments. They demonstrate that biomass burning emissions are a significant contributor to aerosol organosulfates in northern cities, rather than fossil fuel combustion emissions. The overall results provide valuable insights into the formation of aerosol organosulfates associated with biomass burning, making this a noteworthy and meaningful finding. Generally, the manuscript is well-structured and presents a robust experimental approach with clear results. I recommend that this paper be published in Atmospheric Chemistry and Physics once the authors address the following comments.*

Response: We sincerely appreciate your professional and constructive review of our manuscript. Your valuable feedback has greatly improved the clarity and quality of our work. We have carefully revised the manuscript to address the comments.

Major comments:

*1) The identification of over 100 organosulfates is impressive. While the use of*

*surrogate standards is not ideal, it is currently the best solution in the absence of authentic standards. Therefore, please provide a detailed explanation of the criteria used for selecting surrogate standards for quantifying organosulfate species in this section or supplementary information. Additionally, the recoveries of individual organosulfate surrogate standards should be included in the manuscript or supplementary information, as this is crucial for ensuring data quality.*

Response: We are grateful for the insightful comments provided by the reviewer. More discussions have been added in the revised manuscript. Briefly, we further emphasized that the differential ionization efficiencies and fragmentation patterns in the OS measurement may introduce biases (Wang et al. 2017; Wang et al. 2021b). Detailed quantification method and data quality control have also been shown in our previous studies (Wang et al. 2021b; Yang et al. 2023; Yang et al. 2024).

**Supporting Information:**

**S2. Quantification of OSs**

…It is evident that OSs with similar carbon backbone structures typically exhibit analogous MS responses (Wang et al. 2021a). Consequently, the selection of a surrogate standard for a specific OS was predominantly contingent on the similarity between the carbon chain structures of the targeted OS species and the OS standard (Hettiyadura et al. 2017). Furthermore, the sulfur-containing fragment ions observed in the MS/MS spectra of the standard and targeted OS species have been taken into consideration (Hettiyadura et al. 2019; Bryant et al. 2021). The recoveries of the aforementioned surrogate standards were, in order, 88%, 84%, 94%, 89%, 88%, 87%, and 84%. Additional details on the identification of OS compounds, their classification and quantifacation, and data quality control are available in our recent publications (Yang et al. 2023; Yang et al. 2024)…

Additional Comments:

1) *Keywords: I recommend adding "biomass burning" to the list of keywords.*

Response: Thank you for your comment. The revision has been made.

Line 42: …Spatial variation, Influencing factors, Biomass burning…

2) Line 140: The rationale for centrifuging is unclear, especially since a syringe filter was used earlier. It would be helpful to clarify whether the centrifuge was employed later due to solid precipitate formation in the extracts after adding water.

Response: Indeed, centrifugation does not cause any precipitation. However, the instrument administrator requires us to centrifuge to minimize the risk of instrument blockage.

3) Table S5 and Figure S5: Please describe the methodology used to calculate the relative intensity.

Response: Thank you for your comment. The relative signal intensity refers to the percentage of the target OS signal intensity in the total signal intensity of the OS group to which the target OS belongs.

SI: Table S5: Relative signal intensity of identified $OS_a$ in different smoke particle samples. The relative signal intensity refers to the percentage of the target OS signal intensity in the total signal intensity of the OS group to which the target OS belongs.

Figure S6. Mean relative signal intensities of typical aromatic OSs (i.e., $C_6H_5O_4S^-$, $C_7H_7O_4S^-$, $C_8H_7O_4S^-$, and $C_9H_9O_4S^-$) in different smoke particle samples. The relative signal intensity refers to the percentage of the target OS signal intensity in the total signal intensity of the OS group to which the target OS belongs.

4) Line 392: In the caption of Figure S6, please clarify what panels a and b represent.

*Also, indicate which data were used for the correlation analysis in Figure S8.*

Response: Thank you for your comment. In the caption of Figure S6, we have clarified that panels (a) and (b) represent the Pearson correlation diagrams for the cases in southern cities and northern cities, respectively. Additionally, we have indicated that data from four cities were used for the correlation analysis in Figure S8.

**Supporting Information:**

**Figure S6**. Diagrams presenting Pearson correlations among different OSs and important parameters for the cases in (a) southern cities and (b) northern cities…

**Figure S8**. Diagrams presenting Pearson correlations among the different OSs and important parameters (using data from four cities) ...

5) *It would be beneficial to determine whether the authors can provide a quantitative estimation of the contribution of biomass combustion emissions to aerosol isoprene-derived organosulfates, even though this may be a challenging task.*

Response: We appreciate the valuable comment. On average, biomass burning-related $OS_i$ ($OS_i$-BB) accounted for 58% – 64% and 86% – 87% of the total $OS_i$ concentration in southern and northern cities, respectively. It is imperative to acknowledge that $OS_i$-BB can originate not only from biomass combustion but also from the secondary formation of isoprene emitted from biogenic sources. At least in this study, the higher proportion of the $OS_i$ in northern cities can support our consideration that non-biogenic $OS_i$ was an important contributor to $OS_i$ in northern cities. However, given the potential for both biomass burning and biogenic isoprene to contribute to $OS_i$ formation, separating their respective contributions remains challenging, particularly when relying solely on $OS_i$ concentrations. To disentangle the contribution of biomass combustion emissions to $OS_i$, further detailed studies are

necessary.

Lines 445–447: …Given the potential for both biomass burning and biogenic isoprene to contribute to $OS_i$ formation, separating their respective contributions remains challenging…

**Figure 5** …It is noteworthy that $OS_i$-BB can originate not only from biomass combustion, but also from the secondary formation of isoprene emitted from biogenic sources…

Technical Corrections:

1) *Line 146: Change "an optimized solution" to "the optimized solution."*

Response: Thank you for your suggestion. The sentence has been rephrased in the revised manuscript.

Lines 142-144: …we also acknowledge that the developed hydrophilic interaction liquid chromatography method may provide another solution for the measurement of low-MW OSs…

2) *Lines 149 and 153: Please correct the two spelling errors present.*

Response: Thank you for your careful review. The revision has been made.

Line 146: …In addition, it has been indicated in previous studies…

Line 150: …Moreover, there is currently no study evaluating the relative…

3) *Line 364: Amend "... be also supported…" to "...also be supported…"*

Response: The revision has been made.

Line 366: …This can also be supported by previous principal component analysis…

4) *S2. Quantification of OSs. Line 116: Ensure the correct citation format: (Hettiya… Ding et al. 2022a).*

Response: We have updated the citation.

**Supporting information**

…Consequently, the majority of the identified OSs were quantified using surrogate standards (Hettiyadura et al. 2019; Bryant et al. 2021; Wang et al. 2018; Ding et al. 2022)…

5) *Figures S3 and S5: I suggest removing the hyphen in "Ali.-OSs" and "Aro.-OSs."*

Response: Thank you for your kind suggestion to improve the clarity of the figures. Although the hyphen was retained, we re-output the figure to make the display of each parameter clearer.

**Supporting information**

[Figure]

**Figure S3.** Diagrams presenting Pearson correlations among the concentrations of $O_x$, $SO_2$, $SO_4^{2-}$, and the different OSs (using data from four cities). The numbers in the matrix refer to the correlation coefficients ($r$). Symbols * and ** indicate $P < 0.05$ and $P < 0.01$, respectively.

[Figure]

**Figure S6**. Diagrams presenting Pearson correlations among different OSs and important parameters for the cases in (a) southern cities and (b) northern cities. The numbers in the matrix refer to the correlation coefficients ($r$). Symbols * and ** indicate $P < 0.05$ and $P < 0.01$, respectively.

**At last, we deeply appreciate the time and effort you've spent in reviewing our manuscript.**

**Reference:**

Bryant, D. J., Elzein, A., Newland, M., White, E., Swift, S., Watkins, A., Deng, W., Song, W., Wang, S., Zhang, Y., Wang, X., Rickard, A. R., and Hamilton, J. F.: Importance of Oxidants and Temperature in the Formation of Biogenic Organosulfates and Nitrooxy Organosulfates, ACS Earth and Space Chemistry, 5, 2291-2306, 10.1021/acsearthspacechem.1c00204, 2021.

Ding, S., Chen, Y., Devineni, S. R., Pavuluri, C. M., and Li, X.-D.: Distribution characteristics of organosulfates (OSs) in PM2.5 in Tianjin, Northern China: Quantitative analysis of total and three OS species, Sci. Total Environ., 834, 10.1016/j.scitotenv.2022.155314, 2022.

Hettiyadura, A. P. S., Al-Naiema, I. M., Hughes, D. D., Fang, T., and Stone, E. A.: Organosulfates in Atlanta, Georgia: anthropogenic influences on biogenic secondary organic aerosol formation, Atmos. Chem. Phys., 19, 3191-3206, 10.5194/acp-19-3191-2019, 2019.

Hettiyadura, A. P. S., Jayarathne, T., Baumann, K., Goldstein, A. H., de Gouw, J. A., Koss, A., Keutsch, F. N., Skog, K., and Stone, E. A.: Qualitative and quantitative analysis of atmospheric organosulfates in Centreville, Alabama, Atmos. Chem. Phys., 17, 1343-1359, 10.5194/acp-17-1343-2017, 2017.

Wang, Y., Ren, J., Huang, X. H. H., Tong, R., and Yu, J. Z.: Synthesis of Four Monoterpene-Derived Organosulfates and Their Quantification in Atmospheric Aerosol Samples, Environ. Sci. Technol., 51, 6791-6801, 10.1021/acs.est.7b01179, 2017.

Wang, Y., Zhao, Y., Wang, Y., Yu, J.-Z., Shao, J., Liu, P., Zhu, W., Cheng, Z., Li, Z., Yan, N., and Xiao, H.: Organosulfates in atmospheric aerosols in Shanghai, China: seasonal and interannual variability, origin, and formation mechanisms, Atmos. Chem. Phys., 21, 2959-2980, 10.5194/acp-21-2959-2021, 2021a.

Wang, Y., Zhao, Y., Wang, Y., Yu, J. Z., Shao, J., Liu, P., Zhu, W., Cheng, Z., Li, Z., Yan, N., and Xiao, H.: Organosulfates in atmospheric aerosols in Shanghai, China: seasonal and interannual variability, origin, and formation mechanisms, Atmos. Chem. Phys., 21, 2959-2980, 10.5194/acp-21-2959-2021, 2021b.

Wang, Y., Hu, M., Guo, S., Wang, Y., Zheng, J., Yang, Y., Zhu, W., Tang, R., Li, X., Liu, Y., Le Breton, M., Du, Z., Shang, D., Wu, Y., Wu, Z., Song, Y., Lou, S., Hallquist, M., and Yu, J.: The secondary formation of organosulfates under interactions between biogenic emissions and anthropogenic pollutants in summer in Beijing, Atmos. Chem. Phys., 18, 10693-10713, 10.5194/acp-18-10693-2018, 2018.

Yang, T., Xu, Y., Ma, Y.-J., Wang, Y.-C., Yu, J. Z., Sun, Q.-B., Xiao, H.-W., Xiao, H.-Y., and Liu, C.-Q.: Field Evidence for Constraints of Nearly Dry and Weakly Acidic Aerosol Conditions on the Formation of Organosulfates, Environmental Science & Technology Letters, 10.1021/acs.estlett.4c00522, 2024.

Yang, T., Xu, Y., Ye, Q., Ma, Y.-J., Wang, Y.-C., Yu, J.-Z., Duan, Y.-S., Li, C.-X., Xiao, H.-W., Li, Z.-Y., Zhao, Y., and Xiao, H.-Y.: Spatial and diurnal variations of aerosol organosulfates in summertime Shanghai, China: potential influence of photochemical processes and anthropogenic sulfate pollution, Atmos. Chem. Phys., 23, 13433-13450, 10.5194/acp-23-13433-2023, 2023.